# Changes in the Presentation and Severity of Acute Appendicitis: A Comparison of the COVID-19 Pandemic and Post-Pandemic Eras

**DOI:** 10.3390/diseases12110270

**Published:** 2024-11-01

**Authors:** Mohammed Bu Bshait, Ahmed Kamal, Abdullah Almaqhawi, Ahmad Al Abdulqader, Haytham Alarfaj, Mohammed Albarqi, Nawaf Al Khashram, Nora AlMssallem, Fatimah Aljalal, Sara Aljaafari, Abtesam Alnaim, Sharifah Alzabdawi, Ahmad Odeh

**Affiliations:** 1Departments of Surgery, College of Medicine, King Faisal University, Al Hofuf 31982, Saudi Arabiaa.alabdulqader@kfu.edu.sa (A.A.A.); hmarfaj@kfu.edu.sa (H.A.); 2Department of Family and Community Medicine, College of Medicine, King Faisal University, Al Hofuf 31982, Saudi Arabia; 3Department of Biomedical Sciences, College of Medicine, King Faisal University, Al Hofuf 31982, Saudi Arabia; 4College of Medicine, King Faisal University, Al Hofuf 31982, Saudi Arabia; 5Department of Surgery, Prince Saud Bin Jalawi Hospital, Alahsa 36377, Saudi Arabia

**Keywords:** acute appendicitis, complicated appendicitis, COVID-19, pandemic, post-pandemic

## Abstract

Background: The restrictions during the COVID-19 pandemic, along with people’s avoidance of hospital exposure, led to a reduction in medical consultations and delays in care seeking. Numerous reports have shown a decrease in the number of appendectomies performed and an increased incidence of complicated appendicitis during the pandemic. We aimed to investigate these findings during and after the cessation of COVID-19-related restrictions. Methods: This retrospective cohort study was conducted in a single centre, including consecutive adult patients who underwent appendectomy for acute appendicitis during three different periods: the pandemic cohort (7 March–30 June 2020), the post-pandemic cohort (7 March–30 June 2022) and the pre-pandemic control cohort (7 March–30 June 2019). A total of 103 patients were included and divided into the three cohorts. The patients’ demographics, clinical presentation, investigations, and operative data were compared. Results: The three groups did not differ significantly regarding demographics, clinical characteristics, or the number of appendectomies. However, a duration of symptoms at presentation of less than 2 days was significantly more frequent in the pre-pandemic and post-pandemic groups than the pandemic group (*p* = 0.001). The interval between admission and operation was significantly shorter in the post-pandemic group than in the pandemic group. The pandemic group also showed a higher incidence of complicated appendicitis compared to the others (*p* = 0.025). Conclusions: The termination of the COVID-19 pandemic and its related restrictions promoted the earlier presentation of acute appendicitis cases and lowered the incidence of complicated appendicitis. This emphasises the impact of the COVID-19 pandemic on acute appendicitis.

## 1. Introduction

Coronavirus disease 2019 (COVID-19) has posed a serious threat to human life [1,2]. It has caused nearly seven million deaths worldwide since its emergence [3]. The lack of background on the disease and its rapid spread across the world posed significant challenges in the early phase of the pandemic [4]. Authorities prioritised the preservation of life and disease containment [5], which required prepared healthcare systems and well-established prevention strategies.

To address the rapid increase in COVID-19 cases, vigorous precautionary measures were implemented, like border closures, social distancing, and calls to stay at home [6]. However, the COVID-19 pandemic was difficult to control and put healthcare systems under immense pressure. The accelerated propagation of COVID-19 overwhelmed healthcare systems and led to the prioritisation and reservation of resources to face the pandemic [7,8,9] at the cost of other medical services [10,11,12,13].

Although people tended to refrain from visiting hospitals to avoid contracting COVID-19 during the pandemic [14,15], the restrictive measures, like lockdowns and curfews, also limited the accessibility of healthcare facilities when needed [16]. These factors were reflected in reports showing a remarkable decline in emergency department visits, a reduction in surgical emergency admissions, and more-advanced disease stages at presentation [17,18,19].

For acute appendicitis (AA), timely diagnosis and intervention are essential to promote positive outcomes and prevent complications [20,21]. The fulfilment of these requirements could have been altered during the pandemic due to resource allocation and hospital accessibility issues, as well as patients’ avoidance of hospital exposure [22]. Various studies have examined the impact of the COVID-19 pandemic on AA. During the pandemic, a decreased number of appendectomies and a higher rate of complicated appendicitis were observed [23,24,25,26]. However, these variations in AA have not been explored in the post-pandemic phase.

In Saudi Arabia, the first case of COVID-19 was reported on 2 March 2020. One week later, authorities implemented a lockdown to hold back the disease, with health facilities being readied [27]. This may have prevented people from seeking medical attention for issues other than COVID-19. However, the responsible authorities established alternative pathways for such issues, including visual clinics, hotlines for medical consultations, and roaming permits for those who required medical care [28].

By the end of June 2020, the lockdown and curfew restrictions were lifted. However, COVID-19 diagnosis and tracing continued, along with precautionary measures. The easing of restrictions continued as the knowledge of COVID-19 patterns expanded, along with the recovery of healthcare systems and vaccine endorsement [28]. In Saudi Arabia, inoculation against COVID-19 was initiated in December 2020. In March 2022, Saudi Arabia declared the end of the COVID-19 pandemic and related restrictions [29].

Given the transition from the pandemic to post-pandemic era, we aimed to analyse the changes in AA presentation and severity during the COVID-19 pandemic and following the discontinuation of pandemic procedures.

## 2. Materials and Methods

### 2.1. Study Design

This was a retrospective cohort study conducted at a single centre in Saudi Arabia to assess the pattern of AA presentation and severity in different cohorts in relation to COVID-19 phases. Adult patients aged 18 years or above who underwent an appendectomy for AA at Prince Saud bin Jalawi Hospital, Al-Ahsa, Saudi Arabia, were included in their respective cohorts. The pandemic cohort included patients who presented from 7 March to 30 June 2020, during which lockdown measures had been implemented. Patients presenting after the termination of pandemic protocols in Saudi Arabia were included in the post-pandemic cohort from 7 March to 30 June, 2022. The pre-pandemic control cohort covered the corresponding period (between 7 March and 30 June) in 2019.

### 2.2. Data Collection

The medical records of the patients in the study were retrieved. Exclusion criteria included patients undergoing treatment outside the targeted periods, those who were below the age of 18, those who were treated non-surgically, and those who underwent interval appendectomy. A total of 103 patients who underwent an appendectomy for AA were included in the study, with 34 patients in the pre-pandemic group, 31 patients in the pandemic group and 38 patients in the post-pandemic group.

Comprehensive data on demographics, clinical presentations, investigations, and operative records were collected. Complicated appendicitis was determined according to operative and histopathology reports. It included gangrenous appendicitis, the presence of necrosis, perforated appendicitis, and peri-appendicular abscess formation.

### 2.3. Sample Size

A power analysis was conducted using G power software (version 3.1.9.5) to determine the sample size required to run a chi-square test of independence in a 2 (complicated appendicitis, inflamed appendix) × 3 (before, during, and after the pandemic) contingency table with a medium effect size (w = 0.3) (which was adopted from the meta-analysis conducted by Grossi et al. [22]), an alpha level of 0.05, and a power of 80%, indicating that a sample size of approximately 108 participants was required. However, 103 patients were included in the study, which is not far from the calculated sample size.

### 2.4. Study Outcomes

The primary outcomes of this study were the differences in the duration of symptoms at presentation and the incidence of complicated appendicitis between the study cohorts. The secondary outcomes were the differences in clinical characteristics, in-hospital delay, number of appendectomies performed, and operation type.

### 2.5. Statistical Analysis

IBM SPSS version 25 (IBM Corp., Armonk, NY, USA) was used for statistical analysis. Descriptive statistics (numbers and percentages) were presented for all categorical variables, while means and standard deviations (SDs) were used to represent all continuous variables. An exploration of the data using the Shapiro–Wilk and Kolmogorov–Smirnov tests revealed that they did not exhibit a normal distribution. Consequently, nonparametric tests were applied. The Mann–Whitney U test was employed for independent samples to assess associations between continuous and categorical variables. Additionally, the chi-square test (χ^2^) of independence was utilised to examine relationships between categorical variables. Significance was determined at a *p*-value of <0.05 for a 95% confidence interval.

## 3. Results

A total of 106 patients were initially screened. In accordance with the exclusion criteria, three patients were excluded. The reason for exclusion was that they underwent interval appendectomy (two patients from the pre-pandemic cohort and one from the pandemic cohort). The remaining 103 patients were included in the study, of whom 66 were male (64.1%) and 37 were female (35.9%). The mean and SD of patient age were 28.2 ± 10.1 years. The surgeries were distributed across the periods, with 33% occurring pre-pandemic, 30.1% during the pandemic, and 36.9% post-pandemic. Most patients (88.3%) experienced symptoms for less than 2 days, while 11.7% had symptoms for 2 days or more. A significant majority (82.5%) reported nausea or anorexia, whereas 17.5% did not. Vomiting was less common, reported by only 13.6% of patients, compared to 86.4% who did not experience vomiting. Fever was reported in 35% of patients. The mean and SD of white blood cell (WBC) count were 12.2 ± 4.82 *×* 10^9^/L. Computed tomography (CT) was a frequently utilised investigation modality (88.3%). The interval from patient admission to operation was 10.6 ± 5.84 h. The operative findings revealed that 80.6% had an inflamed appendix and 19.4% had complicated appendicitis. The type of operation performed varied, with 62.1% undergoing open surgery and 37.9% having a laparoscopic procedure. The operative times also differed, with 52.4% of surgeries lasting less than 60 min, 25.2% taking between 60 and 120 min, and 22.3% exceeding 120 min (Table 1).

The results of a chi-square test comparing patients who underwent an appendectomy before, during, and after the pandemic revealed interesting findings regarding different variables. The number of appendectomies showed no significant difference between the pre-pandemic, pandemic, and post-pandemic groups, with a chi-square value of 0.718 (df = 2) and a *p*-value of 0.698. There was also no significant difference in the age and gender distributions, with *p*-values of 0.311 and 0.924, respectively. However, the duration of symptoms differed significantly (chi-square value = 13.201, df = 2, *p* = 0.001), indicating that more patients had symptoms for less than 2 days in the pre-pandemic and post-pandemic groups than in the pandemic group. The presence of fever did not vary significantly across the periods (chi-square value = 0.303, df = 2, *p* = 0.860), nor did the presence of nausea or anorexia (chi-square value = 0.342, df = 2, *p* = 0.843) or vomiting (chi-square value = 1.541, df = 2, *p* = 0.463). There was no statistically significant difference in WBC counts or CT scan use between the study groups.

The interval from admission to operation was similar in the pre-pandemic and post-pandemic groups, with mean ± SD values of 7.03 ± 3.01 and 7.95 ± 3.13 *h*, respectively. However, the pandemic group showed a significantly longer interval of 17.6 ± 4.39 *h* (*p* < 0.001). Operative findings also varied significantly (chi-square value = 7.338, df = 2, *p* = 0.025), with a higher occurrence rate of complicated appendicitis during the pandemic. The type of operation performed showed a significant shift (chi-square value = 15.923, df = 1, *p* = 0.000), with more open surgeries conducted during the pandemic and more laparoscopic surgeries post-pandemic. However, the operative duration did not differ significantly (chi-square value = 7.370, df = 2, *p* = 0.118) (Table 2).

## 4. Discussion

The influence of the COVID-19 pandemic on AA has been investigated previously. However, although several studies have demonstrated changes in the course and progression of the disease to its complicated form during the pandemic, the persistence of these variations has not been explored in the post-pandemic setting. In the current study, we aimed to explore these factors in both the pandemic and post-pandemic eras.

Healthcare practices had to adapt to the pandemic’s circumstances. Thus, resources and service capabilities were redirected to fight the outbreak. In the surgical field, this was reflected in recommendations to cancel elective surgeries and the application of conservative management whenever feasible [30,31]. Despite being controversial, the latter approach was proposed for uncomplicated appendicitis during the COVID-19 pandemic [31]. However, the uncertainty of its success made the decision practically challenging, and careful consideration was required [32,33].

Many studies showed a decreased number of appendectomies during the pandemic. They correlated this finding with the potential rise in conservative treatment or the resolution of mild and self-treated AA by patients at home [23,24,34,35,36]. However, we found no significant change in the number of appendectomies performed during the pandemic compared to the pre- and post-pandemic periods. This could be due to the great effort of the authorities to guide the population regarding COVID-19- and non-COVID-19-related concerns.

The duration of symptoms in AA is a factor presumed to influence its course and outcome. During the COVID-19 pandemic, several studies observed a notable increase in symptom duration at the time of AA presentation [23,26,37,38]. Lockdown measures, instructions to avoid public venues, and the fear of contracting COVID-19 were suggested as contributing factors. Similarly, this observation was evident in the current study. We found that nearly 30% of the pandemic group presented 2 days or more from symptom initiation. This time frame is linked with an increased risk of complicated appendicitis. According to Jiang Li et al. [39], patients with AA presenting 2 days or more following symptom onset had a four-fold increase in perforation risk. On the other hand, the delay observed in the pandemic group was neutralised in the post-pandemic group, where significantly earlier presentation of AA compared to the pandemic group was found. This supports the idea that COVID-19 containment, the lifting of related restrictions, and successful vaccination campaigns eliminated the effect of the pandemic on AA presentation.

Our study revealed a significant difference in the interval from admission to operation between groups, with a shorter interval in the post-pandemic group. This was apparently due to the discontinuation of the policies applied during the pandemic, which included COVID-19 testing. The test results are only obtained after 6–12 h, extending the time until intervention. Comparable results have been previously reported [40,41].

In ordinary circumstances, complicated appendicitis comprised about 4–18% of AA cases [42]. The current study revealed an incidence of 11.8% and 13.2% in pre-pandemic and post-pandemic patients, respectively. However, this incidence rate increased significantly to 35.5% in the pandemic group (*p* = 0.025). This finding is consistent with previously published data [38,40,43]. Since it was a publicly common belief that hospitals were a vulnerable place to contract COVID-19, we hypothesize that patients’ fear and reluctance to seek medical care was an influencing factor regarding this observation. Furthermore, the adoption of preoperative COVID-19 screening during the pandemic added further delay till intervention and subsequently increased the risk of complicated appendicitis. However, we suppose that the curfew was of limited impact in our case since roaming permits to seek medical care were readily and immediately available through digital application [28]. Notably, the incidence of complicated appendicitis normalised post-pandemic. This further confirms the negative effect of the COVID-19 pandemic on AA cases.

Interestingly, our data revealed a decline in laparoscopic appendectomies during the pandemic. This is probably due to doubts regarding aerosol contamination and concerns about COVID-19 transmission in laparoscopic procedures [44]. However, with the end of the pandemic, in accordance with supporting evidence [45], laparoscopic intervention was resumed ordinarily.

The limitations of this study include its retrospective nature, small sample size, and the absence of conservatively managed patients. Moreover, it was a single-centre study, which may limit the generalizability of the study findings. However, the study highlights the influence of emerging health system protocols and patients’ healthcare-seeking behaviours in common surgical emergencies under certain circumstances. These factors may be considered by stakeholders to minimise collateral impacts during such events in the future.

## 5. Conclusions

Following the containment of the outbreak and successful vaccination campaigns, the termination of the COVID-19 pandemic and its related restrictions promoted the earlier presentation of AA and lowered the incidence of complicated appendicitis. These findings emphasise the adverse impacts of the COVID-19 pandemic on AA cases.

## Figures and Tables

**Table 1 diseases-12-00270-t001:** Patient demographic and clinical characteristics (*n* = 103).

Variables		N (%)
Gender	Male	66 (64.1%)
	Female	37 (35.9%)
Group	Pre-pandemic	34 (33.0%)
	Pandemic	31 (30.1%)
	Post-pandemic	38 (36.9%)
Duration of symptoms	Less than two days	91 (88.3%)
	Two days or more	12 (11.7%)
Nausea/anorexia	Yes	85 (82.5%)
	No	18 (17.5%)
Vomiting	Yes	14 (13.6%)
	No	89 (86.4%)
Presence of fever	Yes	36 (35%)
	No	67 (65%)
CT	Conducted	91 (88.3%)
	Not conducted	12 (11.7%)
Operative findings	Inflamed appendix	83 (80.6%)
	Complicated appendicitis	20 (19.4%)
	Normal appendix	0 (0.0%)
Operative type	Open	64 (62.1%)
	Laparoscopic	39 (37.9%)
Operative time	<60 min	54 (52.4%)
	60–120 min	26 (25.2%)
	>120 min	23 (22.3%)
	Mean ± SD	
Age	28.2 ± 10.1	
WBC	12.2 ± 4.82	
Interval from admission to operation (hours)	10.6 ± 5.84	

**Table 2 diseases-12-00270-t002:** Displays the cross-tabulation along with the outcomes of the chi-square test pertaining to patients who underwent an appendectomy before, during, and after the pandemic.

Factor		Pre-Pandemic N (%)	Pandemic N (%)	Post-Pandemic N (%)	X^2^ (df)	*p*-Value
Number of appendectomies		34 (33.0%)	31 (30.1%)	38 (36.9%)	0.718(2)	0.698
Gender	Male	22 (64.7%)	19 (61.3%)	25 (65.8%)	0.159(2)	0.924
	Female	12 (35.7%)	12 (38.7%)	13 (34.2%)		
Duration of symptoms	Less than two days	32 (94.1%)	22 (71%)	37 (97.4%)	13.201 (2)	0.001 *
	Two days or more	2 (5.9%)	9 (29%)	1 (2.6%)		
Nausea/anorexia	Yes	27 (79.4%)	26 (74.3%)	32 (84.2%)	0.342(2)	0.843
	No	7 (20.6%)	5 (25.7%)	6 (15.8%)		
Vomiting	Yes	4 (11.8%)	5 (16.1%)	5 (13.2%)	1.541(2)	0.463
	No	30 (88.2)	26 (83.9%)	33 (86.8%)		
Presence of fever	Yes	11 (32.4%)	12 (38.7%)	13 (34.2%)	0.303(2)	0.860
	No	23 (67.6%)	19 (61.3%)	25 (65.8%)		
CT	Conducted	29 (85.3%)	27 (87.1%)	35 (92.1%)	0.596(2)	0.641
	Not conducted	05 (14.7%)	04 (12.9%)	03 (07.9%)		
Operative findings	Inflamed Appendix	30 (88.2%)	20 (64.5%)	33 (86.8%)	7.338(2)	0.025 *
	Complicated appendix	4 (11.8%)	11 (35.5%)	5 (13.2%)		
Operation type	Open	19 (55.9%)	28 (90.3%)	17 (44.7%)	15.923(1)	0.000 *
	Laparoscopic	15 (44.1%)	3 (9.7%)	21 (55.3%)		
Operative time	<60 min	18 (52.9%)	11 (35.5%)	25 (65.8%)	7.370(2)	0.118
	60–120 min	7 (20.6%)	12 (38.7%)	7 (18.4%)		
	>120 min	9 (26.5%)	8 (25.8%)	6 (15.8%)		
		Pre-pandemic Mean ± SD (n = 34)	Pandemic Mean ± SD (n = 31)	Post-pandemic Mean ± SD (n = 38)		*p*-value
Age		26.3 ± 7.9	27.9 ± 9.65	24.7 ± 6.77		0.311
WBC		12.3 ± 4.80	11.9 ± 5.52	12.4 ± 4.32		0.845
Interval from admission to operation (hours)		7.03 ± 3.01	17.6 ± 4.39	7.95 ± 3.13		<0.001 *

X^2^ person chi-square, * significant at *p* < 0.05 level, N = 103.

## Data Availability

The data presented in this study are available on request from the corresponding author due to their containing information that could compromise the participants privacy.

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
