# Peer review of "Changes in the Presentation and Severity of Acute Appendicitis: A Comparison of the COVID-19 Pandemic and Post-Pandemic Eras"

_diseases, 2024, doi:10.3390/diseases12110270_

Round 1

Reviewer 1 Report

Comments and Suggestions for Authors

The manuscript investigates the impact of the COVID-19 pandemic on the presentation and severity of acute appendicitis (AA) by comparing data from pre-pandemic, pandemic, and post-pandemic periods. This retrospective cohort study, conducted at a single center in Saudi Arabia, aims to explore how the global health crisis influenced the clinical management of AA, particularly focusing on symptom duration, time to surgery, and the incidence of complicated appendicitis.

The authors have done an excellent job addressing the concerns raised during the initial review. The correction of the COVID-19 mortality data to reflect the most current statistics from the WHO enhances the manuscript’s accuracy. The inclusion of additional details in the results section regarding the number of patients screened, excluded, and reasons for exclusion significantly improves the transparency and reproducibility of the study. Clarifying the primary and secondary outcomes in the methods section aligns the study’s objectives with its title, making the research focus clearer. The addition of a power analysis demonstrates the authors' commitment to ensuring the statistical robustness of their findings, despite the acknowledged small sample size. The discussion now includes a well-reasoned hypothesis on the observed increase in complicated appendicitis during the pandemic, providing a more comprehensive interpretation of the results. Finally, the acknowledgment of the study’s single-center nature as a limitation is a valuable addition that appropriately frames the generalizability of the findings.

These revisions have significantly strengthened the manuscript, ensuring that it is both scientifically rigorous and clearly presented. The authors' responsiveness to the feedback provided is commendable, and the revised manuscript is now more informative, accurate, and comprehensive.

Author Response

Open Review (x) I would not like to sign my review report
( ) I would like to sign my review report Quality of English Language ( ) I am not qualified to assess the quality of English in this paper.
( ) The English is very difficult to understand/incomprehensible.
( ) Extensive editing of English language required.
( ) Moderate editing of English language required.
( ) Minor editing of English language required.
(x) English language fine. No issues detected.            

  Yes Can be improved Must be improved Not applicable
Does the introduction provide sufficient background and include all relevant references? (x) ( ) ( ) ( )
Is the research design appropriate? (x) ( ) ( ) ( )
Are the methods adequately described? (x) ( ) ( ) ( )
Are the results clearly presented? (x) ( ) ( ) ( )
Are the conclusions supported by the results? (x) ( ) ( ) ( )

The manuscript investigates the impact of the COVID-19 pandemic on the presentation and severity of acute appendicitis (AA) by comparing data from pre-pandemic, pandemic, and post-pandemic periods. This retrospective cohort study, conducted at a single center in Saudi Arabia, aims to explore how the global health crisis influenced the clinical management of AA, particularly focusing on symptom duration, time to surgery, and the incidence of complicated appendicitis.

The authors have done an excellent job addressing the concerns raised during the initial review. The correction of the COVID-19 mortality data to reflect the most current statistics from the WHO enhances the manuscript’s accuracy. The inclusion of additional details in the results section regarding the number of patients screened, excluded, and reasons for exclusion significantly improves the transparency and reproducibility of the study. Clarifying the primary and secondary outcomes in the methods section aligns the study’s objectives with its title, making the research focus clearer. The addition of a power analysis demonstrates the authors' commitment to ensuring the statistical robustness of their findings, despite the acknowledged small sample size. The discussion now includes a well-reasoned hypothesis on the observed increase in complicated appendicitis during the pandemic, providing a more comprehensive interpretation of the results. Finally, the acknowledgment of the study’s single-center nature as a limitation is a valuable addition that appropriately frames the generalizability of the findings.

These revisions have significantly strengthened the manuscript, ensuring that it is both scientifically rigorous and clearly presented. The authors' responsiveness to the feedback provided is commendable, and the revised manuscript is now more informative, accurate, and comprehensive.

Response: We are so grateful for the reviewer's valuable input that enhanced our manuscript and we are happy about the reviewer's satisfaction of the current version of the manuscript. 

Reviewer 2 Report

Comments and Suggestions for Authors

Accepted the revision

Comments on the Quality of English Language

accept the revision

Author Response

Open Review (x) I would not like to sign my review report
( ) I would like to sign my review report Quality of English Language ( ) I am not qualified to assess the quality of English in this paper.
( ) The English is very difficult to understand/incomprehensible.
( ) Extensive editing of English language required.
( ) Moderate editing of English language required.
(x) Minor editing of English language required.
( ) English language fine. No issues detected.            
  Yes Can be improved Must be improved Not applicable
Does the introduction provide sufficient background and include all relevant references? (x) ( ) ( ) ( )
Is the research design appropriate? ( ) (x) ( ) ( )
Are the methods adequately described? (x) ( ) ( ) ( )
Are the results clearly presented? (x) ( ) ( ) ( )
Are the conclusions supported by the results? ( ) (x) ( ) ( )
    Comments and Suggestions for Authors

Accepted the revision

Comments on the Quality of English Language

accept the revision

Response: We would like to thank the reviewer for the time and effort reviewing our manuscript and we are glad about the reviewer's decision

Reviewer 3 Report

Comments and Suggestions for Authors

The study was well designed, the manuscript is very well organized, and clearly written. The results and the conclusions support studies previously published. 

Author Response

Open Review (x) I would not like to sign my review report
( ) I would like to sign my review report Quality of English Language ( ) I am not qualified to assess the quality of English in this paper.
( ) The English is very difficult to understand/incomprehensible.
( ) Extensive editing of English language required.
( ) Moderate editing of English language required.
( ) Minor editing of English language required.
(x) English language fine. No issues detected.            
  Yes Can be improved Must be improved Not applicable
Does the introduction provide sufficient background and include all relevant references? (x) ( ) ( ) ( )
Is the research design appropriate? (x) ( ) ( ) ( )
Are the methods adequately described? (x) ( ) ( ) ( )
Are the results clearly presented? (x) ( ) ( ) ( )
Are the conclusions supported by the results? (x) ( ) ( ) ( )
    Comments and Suggestions for Authors

The study was well designed, the manuscript is very well organized, and clearly written. The results and the conclusions support studies previously published. 

Response:  We are delighted to receive this feedback about our work. The reviewer's time and effort to review the manuscript are really appreciated. 

Reviewer 4 Report

Comments and Suggestions for Authors

This manuscript shows the effect of the COVID-19 pandemic restrictions and related medical system protocol on the management of patients with acute appendicitis. Though the sample size is lower than the normal clinical record studies and the records are confined in a single centre, the health care seeking approach of the patients and the behaviour of the treatment given by the healthcare provider has almost been the same worldwide. So, it is a good indicative study to see the social and pandemic related treatment regime given to other health conditions which may be dealt later. The comparison of the results between the time dependent groups are showing the same. The data and the statistical analyses have been implemented aptly. But all the aspects covered in this manuscript mainly concerned with the social management and the behaviour of patients and healthcare system protocol, it won’t be able to add more to the data what has already been found and discussed last couple of years. This situation is also not unique to the acute appendicitis cases only.

Could you please show something unique data related to the acute appendix case studies pre and post pandemic era? 

It would have been better to discuss the probable direct effect of the pathophysiology of COVID-19 on the acute appendix, if any. It would add some beneficial aspect to this manuscript.

Author Response

Open Review

Quality of English Language

( ) I am not qualified to assess the quality of English in this paper.
( ) The English is very difficult to understand/incomprehensible.
( ) Extensive editing of English language required.
( ) Moderate editing of English language required.
( ) Minor editing of English language required.
(x) English language fine. No issues detected.

  Yes Can be improved Must be improved Not applicable
Does the introduction provide sufficient background and include all relevant references? (x) ( ) ( ) ( )
Is the research design appropriate? ( ) (x) ( ) ( )
Are the methods adequately described? (x) ( ) ( ) ( )
Are the results clearly presented? (x) ( ) ( ) ( )
Are the conclusions supported by the results? (x) ( ) ( ) ( )

Comments and Suggestions for Authors

This manuscript shows the effect of the COVID-19 pandemic restrictions and related medical system protocol on the management of patients with acute appendicitis. Though the sample size is lower than the normal clinical record studies and the records are confined in a single centre, the health care seeking approach of the patients and the behaviour of the treatment given by the healthcare provider has almost been the same worldwide. So, it is a good indicative study to see the social and pandemic related treatment regime given to other health conditions which may be dealt later. The comparison of the results between the time dependent groups are showing the same. The data and the statistical analyses have been implemented aptly. But all the aspects covered in this manuscript mainly concerned with the social management and the behaviour of patients and healthcare system protocol, it won’t be able to add more to the data what has already been found and discussed last couple of years. This situation is also not unique to the acute appendicitis cases only.

Response 1: We thank the reviewer for pointing this out. We selected acute appendicitis in our study because of its high prevalence requiring a constant attention regardless the surrounding circumstances. 

Could you please show something unique data related to the acute appendix case studies pre and post pandemic era? 

Response 2: Thank you for pointing this out. Available literature have been investigating the pandemic impact on acute appendicitis with comparison to the pre-pandemic situation. This is why we tried to investigate the post-pandemic era and explore the persistence of the reported findings during the pandemic. 

 It would have been better to discuss the probable direct effect of the pathophysiology of COVID-19 on the acute appendix, if any. It would add some beneficial aspect to this manuscript.

Response 3: Thank you for this insightful suggestion. However, this is not within the scope of our study and would be considered for future research. 

Round 2

Reviewer 4 Report

Comments and Suggestions for Authors

Thanks for replying to the review comments. But, this manuscript is having a scope of further improvement especially considering the pandemic group. Presently, this manuscript is out of scope of this journal in its present form. 

Author Response

Reviewer 4’s comments and response:

  • The comparison of the results between the time dependent groups are showing the same. The data and the statistical analyses have been implemented aptly. But all the aspects covered in this manuscript mainly concerned with the social management and the behavior of patients and healthcare system protocol, it won’t be able to add more to the data what has already been found and discussed last couple of years. This situation is also not unique to the acute appendicitis cases only.
  • Response:

We agree that the impact of COVID-19 pandemic on acute appendicitis has been investigated thoroughly worldwide. However, there is lack of data in such field in our region and pandemic settings. Thus, we aimed to explore the case during and after the pandemic with comparison to the ordinary circumstances (pre-pandemic).

Our findings verify the effect of the pandemic measures on the presentation pattern of acute appendicitis and the disease course. These findings support the results of various report in this subject despite the relative differences in the pandemic settings and the applied measures. Highlighting the contributing factors for such results would facilitate the implementation of guidelines to be followed in similar circumstances. The addition of the current study is by investigating the post-pandemic era. This reassures the shift toward neutral acute appendicitis pattern and characteristics.

  • Could you please show something unique data related to the acute appendix case studies pre and post pandemic era?
  • Response:

Acute appendicitis presentation and incidence of complicated form of the disease were almost similar in pre and post-pandemic cases. This reflects the turn down of the pandemic’s influence in the observed findings regarding the delayed presentation and increased incidence of complicated appendicitis. Although COVID-19 disease has not been eliminated, post-pandemic findings showed the cohabitation with it and the patients’ belief in vaccination efficacy which was reflected in their seeking attitude.

  • It would have been better to discuss the probable direct effect of the pathophysiology of COVID-19 on the acute appendix, if any. It would add some beneficial aspect to this manuscript.
  • Response:

Despite the pandemic effect was not directly related to acute appendicitis pathophysiology, it was evident that acute appendicitis pattern has been altered in that period. The relation of COVID-19 as causative agent for acute appendicitis has been postulated in scattered reported cases. This aspect is still area for further exploration. It is an excellent indicative suggestion. However, it requires different study design to validate this prospect.